# Enhanced Superconducting Critical Parameters in a New High-Entropy Alloy Nb_0.34_Ti_0.33_Zr_0.14_Ta_0.11_Hf_0.08_

**DOI:** 10.3390/ma16175814

**Published:** 2023-08-24

**Authors:** Rafał Idczak, Wojciech Nowak, Bartosz Rusin, Rafał Topolnicki, Tomasz Ossowski, Michał Babij, Adam Pikul

**Affiliations:** 1Institute of Experimental Physics, University of Wrocław, pl. M. Borna 9, 50-204 Wrocław, Poland; wojciech.nowak@uwr.edu.pl (W.N.); 317015@uwr.edu.pl (B.R.); rafal.topolnicki@uwr.edu.pl (R.T.); tomasz.ossowski@uwr.edu.pl (T.O.); 2Institute of Low Temperature and Structural Reaseach, Polish Academy of Sciences, ul. Okólna 2, 50-422 Wrocław, Poland; m.babij@intibs.pl (M.B.); a.pikul@intibs.pl (A.P.); 3Dioscuri Center in Topological Data Analysis, Institute of Mathematics, Polish Academy of Sciences, ul. Śniadeckich 8, 00-656 Warsaw, Poland

**Keywords:** high-entropy alloys, superconductivity, X-ray powder diffraction, scanning transmission electron microscopy, resistivity measurements, magnetic susceptibility measurements, specific heat measurements, Korringa–Kohn–Rostoker method, density functional calculations

## Abstract

The structural and physical properties of the new titanium- and niobium-rich type-A high-entropy alloy (HEA) superconductor Nb${}_{0.34}$Ti${}_{0.33}$Zr${}_{0.14}$Ta${}_{0.11}$Hf${}_{0.08}$ (in at.%) were studied by X-ray powder diffraction, energy dispersive X-ray spectroscopy, magnetization, electrical resistivity, and specific heat measurements. In addition, electronic structure calculations were performed using two complementary methods: the Korringa–Kohn–Rostoker Coherent Potential Approximation (KKR-CPA) and the Projector Augmented Wave (PAW) within Density Functional Theory (DFT). The results obtained indicate that the alloy exhibits type II superconductivity with a critical temperature close to 7.5 K, an intermediate electron–phonon coupling, and an upper critical field of 12.2(1) T. This finding indicates that Nb${}_{0.34}$Ti${}_{0.33}$Zr${}_{0.14}$Ta${}_{0.11}$Hf${}_{0.08}$ has one of the highest upper critical fields among all known HEA superconductors.

## 1. Introduction

High entropy alloys (HEAs) are loosely defined as a homogeneous mixture of five or more elements, each with an atomic content of between 5% and 35% [1,2]. The complexity of HEAs compositions gives rise to a so-called “cocktail effect”, which means that various HEAs have unique physical properties that differ from those expected from the simple rule of mixtures [3,4]. Therefore, HEAs represent one of the most exciting and promising research areas in materials science today.

In 2014, the first superconducting high-entropy alloy Ta${}_{0.34}$Nb${}_{0.33}$Hf${}_{0.08}$Zr${}_{0.14}$Ti${}_{0.11}$ (in at.%) was discovered [5]. According to Sun and Cava [6], this alloy can be classified as a type-A HEA superconductor. HEAs of this type contain only metals on the left side of the transition metal region of the periodic table, crystallize in a body-centered cubic (bcc) lattice, and exhibit superconducting transition temperatures ($T_{c}$) between 4.0 and 9.2 K, while their highest upper critical field ($\mu_{0}H_{c2}$) is close to 12 T [6,7,8,9,10,11,12]. These are the most widely studied HEA superconductors to date. However, it should be noted that most of these alloys contain a large amount of niobium and tantalum, while the titanium concentration is typically less than 20 at.%. Because binary Nb-Ti alloys exhibit much better superconducting properties than Nb-Ta [13], it is plausible to expect that NbTi-based high-entropy alloys will have a higher $T_{c}$ and/or $\mu_{0}H_{c2}$ compared to previously reported type-A HEA superconductors. Moreover, in the case of various ternary Nb-Ti-Hf and Nb-Ti-Ta and quaternary Nb-Ti-Ta-Hf alloys, the highest values of $\mu_{0}H_{c2}$ were observed for the titanium-rich alloys [14]. In particular, for Nb${}_{0.25}$Ta${}_{0.10}$Ti${}_{0.65}$ and Nb${}_{0.225}$Ta${}_{0.150}$Ti${}_{0.625}$, the upper critical field measured at 2 K is close to 15.4 T. For Nb${}_{0.35}$Hf${}_{0.05}$Ti${}_{0.60}$ and Nb${}_{0.25}$Ta${}_{0.10}$Hf${}_{0.05}$Ti${}_{0.60}$, the values of $\mu_{0}H_{c2}$(2 K) are slightly lower and equal to 14.5 T and 15 T, respectively.

This paper reports on the synthesis and detailed study of superconductivity in the titanium- and niobium-rich new type-A HEA superconductor Nb${}_{0.34}$Ti${}_{0.33}$Zr${}_{0.14}$Ta${}_{0.11}$Hf${}_{0.08}$ (in at.%). In particular, the crystal structure, chemical composition, and homogeneity of the prepared material are characterized by X-ray diffraction (XRD) and scanning electron microscopy (SEM). Magnetic, electrical resistivity and specific heat measurements reveal the onset of bulk superconductivity in Nb${}_{0.34}$Ti${}_{0.33}$Zr${}_{0.14}$Ta${}_{0.11}$Hf${}_{0.08}$ at about 7.5 K. In addition, the experimental results are supported by electronic structure calculations, which were performed using two complementary approaches: the Korringa–Kohn–Rostoker Coherent Potential Approximation (KKR-CPA) method [15,16] and the Projector Augmented Wave (PAW) method within the Density Functional Theory (DFT) [17]. The results obtained are discussed and compared with corresponding data which were previously reported for other type-A HEA superconductors [5,6,7,8,9,10,11,12,18].

## 2. Materials and Methods

A polycrystalline sample of Nb${}_{0.34}$Ti${}_{0.33}$Zr${}_{0.14}$Ta${}_{0.11}$Hf${}_{0.08}$ was prepared by conventional arc melting method from high purity metals (of at least 99.9% purity) in the form of slugs and chips. The approximately 1 g ingot was arc-melted at least four times to improve the homogeneity of the alloy. A titanium getter was used prior to each arc-melting fusion in order to capture the residual gases in the chamber. Total weight loss after melting was less than 0.2%.

The crystal structure of the prepared sample was studied by X-ray powder diffraction (XRD) using a PANalytical X’pert Pro diffractometer with Cu Kα radiation. The experimental XRD pattern was analyzed by the Rietveld method implemented in the FullProf software package (version January-2023) [19]. The mean level of internal strains ϵ and the mean grain size *L* were determined by the Williamson–Hall method [20]. The homogeneity and chemical composition of the alloy were verified by energy dispersive X-ray spectroscopy (EDXS) using a FESEM FEI Nova NanoSEM 230 scanning electron microscope equipped with an EDAX Genesis XM4 spectrometer on a polished surface of the cut specimen.

Magnetic properties, electrical resistivity, and specific heat were measured as a function of temperature and applied magnetic field using a Quantum Design Physical Properties Measurement System (PPMS) platform. The shape of the sample used for magnetic properties and electrical resistivity measurements was close to a cuboid with dimensions of 3.5 mm × 1.5 mm × 1 mm. The resistivity was measured using a four-contact method by passing an alternating current through the sample with an amplitude of up to 51 mA and a frequency of 33 Hz. The sample was oriented with the largest side perpendicular to the applied DC magnetic field. The AC susceptibility measurements were performed by applying an AC field with an amplitude of $\mu_{0}H_{\mathrm{ac}}$ = 1 mT and a frequency of 1 kHz. The demagnetization factor N=0.58 was determined using the formula proposed by Prozorov and Kogan for the effective demagnetizing factor of a rectangular cuboid as a function of its dimensions *a*, *b*, and *c* [21]:(1)$$N=\frac{4ab}{4ab+3c\left(a+b\right)}$$

For the specific heat measurements, the sample was additionally cut into 1.5 mm × 1.5 mm × 1.0 mm size. The data were collected using the thermal relaxation technique and the two-tau model. The heat capacity of the sample platform with a very small amount of Apiezon N cryogenic grease was determined and subtracted prior to the sample measurement.

A computational method commonly used to study the electron structure of HEA systems is the Korringa–Kohn–Rostoker Coherent Potential Approximation (KKR-CPA) formalism [15,16,22,23,24]. In the present study, the KKR-CPA method implemented in the AkaiKKR (machikaneyama, version cpa2021v01) package [25,26,27] was used. To construct the muffin-tin crystal potential in the semi-relativistic approach, the Perdew–Burke–Ernzerhof exchange-correlation functional (PBE) was used [28,29,30]. The angular momentum cutoff was set to $l_{\max}$ = 3, and 5216 k-points were used to sample the irreducible part of the Brillouin zone during the self-consistent cycle and density of states calculations. The Atomic Sphere Approximation (ASA) was used in all calculations. The lattice parameter $a_{\mathrm{KKR}-\mathrm{CPA}}$ = 3.39 Å, the bulk modulus $B_{\mathrm{KKR}-\mathrm{CPA}}$ = 125.9 GPa and its derivative $B_{\mathrm{KKR}-\mathrm{CPA}}^{\prime }$ = 4.02 of the crystal Nb${}_{0.34}$Ti${}_{0.33}$Zr${}_{0.14}$Ta${}_{0.11}$Hf${}_{0.08}$ were derived using the third-order Birch–Murnaghan equation of state [31]. The results obtained by the KKR-CPA method were additionally verified by supercell DFT calculations in the plane-wave function basis implemented in the Vienna Ab initio Simulation Package (VASP) [32,33,34]. The electron exchange-correlation interactions were treated at the Generalized Gradient Approximation (GGA) level using the PBE functional form. The electron-ion–core interactions were represented by the Projector Augmented Wave (PAW) potentials [17,35]. A plane-wave basis set with a kinetic energy cutoff of 400 eV was used. Because the supercell approach requires a well-defined position of all constituent elements, three different random atomic configurations were considered. All configurations contain 19 Ti, 18 Nb, 5 Hf, 7 Zr, and 5 Ta atoms to approximate the atomic composition of the studied HEA. The atoms were randomly placed at the nodes of the body-centered cubic 3×3×3 supercell. The lattice parameters, derived using the Birch–Murnaghan third-order equation of state, for all three configurations equal to $a_{\mathrm{PAW}}$ = 3.34 Å and $B_{\mathrm{PAW}}$ = 122.5 GPa, are in good agreement with those obtained by the KKR-CPA method ($a_{\mathrm{KKR}-\mathrm{CPA}}$ = 3.39 Å, $B_{\mathrm{KKR}-\mathrm{CPA}}$ = 125.9 GPa). In both KKR-CPA and PAW calculations, atoms are assumed to occupy high-symmetry positions of the bcc lattice. Their positions were not optimized, and only the size of the unit cell was optimized. On the basis of a simple assessment which is presented in Appendix A, it was assumed that the atomic optimization would have a negligibly small impact on the obtained results.

## 3. Results and Discussion

### 3.1. Phase Formation and Crystal Structure

Figure 1 shows the XRD pattern measured at room temperature for the Nb${}_{0.34}$Ti${}_{0.33}$Zr${}_{0.14}$Ta${}_{0.11}$Hf${}_{0.08}$ sample. As can be seen, the diffraction peaks can be well indexed within a body-centered cubic (bcc) crystal structure (space group Im$\bar{3}$m, W-type structure). No reflections from any secondary phases were observed. Therefore, it can be said that the synthesized alloy is a high-entropy alloy, in which all the constituent elements are randomly incorporated into the bcc crystal structure. The values of the fitting parameters $R_{\mathrm{wp}}$ = 11.6 (the weighted profile R-factor) and $R_{\exp}$ = 5.74 (the expected R-factor), usually used to evaluate the quality of the Rietveld refinement, as well as the difference between the observed and calculated patterns shown in Figure 1 demonstrate the good quality of the obtained results [36].

The lattice parameter *a* = 3.349(1) Å of Nb${}_{0.34}$Ti${}_{0.33}$Zr${}_{0.14}$Ta${}_{0.11}$Hf${}_{0.08}$, determined by the Rietveld method, is in good agreement with the theoretical values $a_{\mathrm{KKR}-\mathrm{CPA}}$ = 3.38 Å and $a_{\mathrm{PAW}}$ = 3.34 Å as well as with the composition-averaged value $a_{\mathrm{th}}$ = 3.353 Å calculated using Vegard’s law [37] and the lattice parameters of the alloying elements in bcc structure: Nb (3.301 Å), Ta (3.303 Å), Ti (3.276 Å), Hf (3.559 Å) and Zr (3.559 Å) [7]. At the same time, *a* = 3.349(1) Å obtained for Ti${}_{0.33}$Nb${}_{0.34}$Hf${}_{0.08}$Zr${}_{0.14}$Ta${}_{0.11}$ can be compared with *a* values reported for other type-A HEA superconductors. In particular, slightly higher *a* values (3.36–3.48 Å) are observed for various Ta-Nb-Hf-Zr-Ti and V-Nb-Hf-Zr-Ti alloys [5,7,11]. However, these HEAs contain a low concentration of titanium ($c_{\mathrm{Ti}}$ < 20 at.%). Because the lattice parameter of titanium is smaller than that of niobium, tantalum, zirconium, and hafnium, this result is reasonable.

The Williamson–Hall method of XRD line profile analysis is applied to estimate the mean grain size *L* and the mean level of internal strains ϵ of the polycrystalline Nb${}_{0.34}$Ti${}_{0.33}$Zr${}_{0.14}$Ta${}_{0.11}$Hf${}_{0.08}$ sample. Using the experimental data obtained from the Rietveld refinement of the measured XRD pattern, one can construct the Williamson–Hall plot, which is shown in the inset of Figure 1. The points obtained follow a straight line according to the equation [20]:(2)$$\beta \cos\left(\theta \right)=\epsilon \sin\left(\theta \right)+\frac{K\lambda }{L},$$
where β is the line broadening at half the maximum intensity given in 2θ units, θ denotes the Bragg angle, λ stands for the X-ray wavelength, and *K* is a dimensionless shape factor. Assuming *K* = 0.9, it was found that *L* = 177(1) nm and ϵ = 2.3(1)%.

### 3.2. Homogeneity and Chemical Composition

Figure 2 shows the SEM micrograph and EDXS elemental mapping obtained for the Nb${}_{0.34}$Ti${}_{0.33}$Zr${}_{0.14}$Ta${}_{0.11}$Hf${}_{0.08}$ sample. A visual inspection of Figure 2 shows that the studied material is a microscopically homogeneous mixture of the five constituent elements on the micrometer scale. There are no obvious impurity phases, nor are there any zones significantly enriched in any of the components. The atomic composition determined by the EDXS analysis (see Appendix A) is almost identical to the nominal one within the experimental accuracy.

### 3.3. Electrical Resistivity

Electrical resistivity measured in the temperature range between 300 and 2 K in magnetic fields between 0 and 9 T is shown in Figure 3. At elevated temperatures, the measured ρ values decrease systematically with temperature indicating a metallic nature of the studied HEA. In particular, ρ = 95 μΩ cm at RT and decreases to ρ = 82 μΩcm at 8 K. The calculated residual resistivity ratio RRR is close to 1.2. The rather small value of RRR is probably related to the polycrystalline nature of the sample as well as to the high degree of atomic disorder that is a common feature of HEAs. Similar results were previously obtained for Ta${}_{0.34}$Nb${}_{0.33}$Hf${}_{0.08}$Zr${}_{0.14}$Ti${}_{0.11}$[5], Hf${}_{0.21}$Nb${}_{0.25}$Ti${}_{0.15}$V${}_{0.15}$Zr${}_{0.24}$[11] and (NbTa)${}_{0.67}$(MoHfW)${}_{0.33}$ [12].

A sharp drop in the resistivity at low temperature manifests the superconducting (SC) transition with the onset critical temperature $T_{c}^{\mathrm{onset}}$ close to 7.8 K. In zero magnetic field, the resistance vanishes at $T_{c}^{\rho 0\%}\approx 7.5$ K. In addition, as can be seen in the inset of Figure 3, the SC transition is systematically shifted to lower temperatures with increasing fields. However, at relatively high magnetic field of $\mu_{0}H$ = 9 T, the SC transition is still observed at $T_{c}^{\mathrm{onset}}\approx 3.7$ and $T_{c}^{\rho 0\%}\approx 3.1$ K. This finding suggests that the studied material may have one of the highest upper critical field among all known HEA superconductors [6,8,10,18].

### 3.4. Magnetic Properties

Temperature dependence of AC susceptibility measured in various magnetic fields is presented in Figure 4. The diamagnetic behavior observed in the real part of the AC susceptibility $\chi^{\prime }$ confirms the existence of bulk superconductivity in Nb${}_{0.34}$Ti${}_{0.33}$Zr${}_{0.14}$Ta${}_{0.11}$Hf${}_{0.08}$. In the case of the imaginary part $\chi^{\prime \prime }$, the experimental curves show peaks that reflect energy dissipation and are characteristic of superconductors. In particular, for type II superconductors, as the temperature increases from below toward $T_{c}$, flux lines and bulk shielding currents begin to penetrate the superconductor when the applied field exceeds the lower critical field. This causes losses of energy, and when the flux lines and shielding currents fully penetrate the material, the losses reach a maximum. As the temperature rises above $T_{c}$, the losses drop to zero [38]. Taking the above into account, one can determine the $T_{c}$ as a function of the applied magnetic field using $\chi^{\prime }\left(T\right)$ and $\chi^{\prime \prime }\left(T\right)$ data. At $\mu_{0}H$ = 0.01 T, $T_{c}$ is 7.37 K, while at $\mu_{0}H$ = 9 T, $T_{c}$ is close to 3 K. These temperatures are slightly lower than $T_{c}^{\mathrm{onset}}$ and comparable with $T_{c}^{\rho 0\%}$ values obtained from the electrical resistivity measurements.

Temperature dependences of DC mass magnetization, σ in various magnetic fields, measured in the zero-field cooling (ZFC) and field cooling (FC) regimes, are shown in Figure 5. The results are in agreement with the AC susceptibility data, indicating the occurrence of bulk superconductivity in the studied HEA below $T_{c}$. At $\mu_{0}H$ = 0.01 T, the onset is at $T_{c}$ = 7.0 K. As can be seen, at $\mu_{0}H$ = 7 and 9 T, the measured σ values are positive down to 2 K. However, the decrease of σ at 4.4 K (in $\mu_{0}H$ = 7 T) and at 3.3 K (in $\mu_{0}H$ = 9 T) is indicative of the onset of superconductivity.

Magnetic field dependences of the DC mass magnetization measured at 2, 4, and 6 K are plotted in Figure 6. The data obtained in the low field region show that Nb${}_{0.34}$Ti${}_{0.33}$Zr${}_{0.14}$Ta${}_{0.11}$Hf${}_{0.08}$ is type II superconductor. Using the demagnetization factor N=0.58, one can estimate the dimensionless volume susceptibility $\chi_{\mathrm{SI}}$ which is close to −0.73 in a magnetic field of $\mu_{0}H$ = 15 mT at 2 K. This value is higher than the expected value of $\chi_{\mathrm{SI}}=-1$ for the fully developed Meissner state. Therefore, it is plausible to assume that for Nb${}_{0.34}$Ti${}_{0.33}$Zr${}_{0.14}$Ta${}_{0.11}$Hf${}_{0.08}$, the lower critical field $\mu_{0}H_{c1}<$ 15 mT at 2 K. This result is comparable to $\mu_{0}H_{c1}\approx$ 32 mT determined for Ta${}_{0.34}$Nb${}_{0.33}$Hf${}_{0.08}$Zr${}_{0.14}$Ti${}_{0.11}$[5] and $\mu_{0}H_{c1}\approx$ 7 mT reported for (NbTa)${}_{0.67}$(MoHfW)${}_{0.33}$[12]. In higher magnetic fields, the magnetization curves behave in the typical manner of the type II superconductor in the vortex state. Unfortunately, the values of $\mu_{0}H_{c2}$ are difficult to determine since the paramagnetic contribution becomes dominant at high fields. However, the detailed analysis of the σ(H) data obtained at 4 and 6 K shows that the difference between these two curves becomes negligibly small and field independent above $\mu_{0}H$ = 7 T. This finding suggests that $\mu_{0}H_{c2}\approx$ 7 T at 4 K. At the same time, the curves obtained at 2 and 4 K are not connected even at $\mu_{0}H$ = 9 T, indicating that $\mu_{0}H_{c2}>$ 9 T at 2 K. These estimates are in good agreement with the values obtained from the temperature dependence of the AC susceptibility and the DC magnetization.

The upper critical fields as a function of temperature derived from the magnetic ($\chi_{\mathrm{AC}}$ and $\sigma_{\mathrm{DC}}$) and resistivity ($\rho^{0\%}$) data are shown in Figure 7. To describe the obtained experimental $\mu_{0}H_{c2}$ data, a full Werthamer–Helfand–Hohenberg (WHH) formalism for isotropic-gap BCS superconductors in the dirty limit, incorporating the spin-paramagnetic effect via the Maki parameter $\alpha_{M}$ and the spin-orbit scattering constant $\lambda_{\mathrm{SO}}$, was used [39,40,41]:(3)$$\ln\frac{1}{t}=\left(\frac{1}{2}+\frac{i\lambda_{SO}}{4\gamma }\right)\psi \left(\frac{1}{2}+\frac{\bar{h}+\frac{1}{2}\lambda_{SO}+i\gamma }{2t}\right)+\left(\frac{1}{2}-\frac{i\lambda_{SO}}{4\gamma }\right)\psi \left(\frac{1}{2}+\frac{\bar{h}+\frac{1}{2}\lambda_{SO}-i\gamma }{2t}\right)-\psi \left(\frac{1}{2}\right),$$
where $t=\frac{T}{T_{c}}$, $\bar{h}=\frac{4}{\pi^{2}}\frac{H_{c2}}{-dH_{c2}/dT}$, $\gamma =\sqrt{{\left(\alpha_{M}\bar{h}\right)}^{2}-{\left(\frac{1}{2}\lambda_{SO}\right)}^{2}}$, *i* denotes an imaginary number and ψ is a digamma function. As can be seen in Figure 7, the best agreement with the experimental data can be obtained using the WHH model with $\alpha_{M}=1.4$ and $\lambda_{\mathrm{SO}}=4.5$, giving $\mu_{0}H_{c2}\left(0\right)=12.2\left(1\right)$ T. Comparing this value with the corresponding data presented in works [6,8,10,18], it can be observed that Nb${}_{0.34}$Ti${}_{0.33}$Zr${}_{0.14}$Ta${}_{0.11}$Hf${}_{0.08}$ has one of the highest upper critical fields in the family of type-A HEA superconductors. Moreover, it should be noted that the values of $\mu_{0}H_{c2}\left(0\right)$ presented in works [6,10,18] were determined using different theoretical functions that do not take into account the effects of Pauli spin paramagnetism and spin–orbit scattering on the temperature dependence of $H_{c2}$ in type II superconductors [40]. Therefore, the $\mu_{0}H_{c2}\left(0\right)$ values presented in works [6,10,18] for various type-A HEAs are highly overestimated. In the case of Ti${}_{0.15}$Zr${}_{0.15}$Nb${}_{0.35}$Ta${}_{0.35}$, the WHH model predicts $\mu_{0}H_{c2}\left(0\right)=11.6$ T [8], which is slightly lower than $\mu_{0}H_{c2}\left(0\right)=12.2\left(1\right)$ T obtained for the HEA studied in this work. Nevertheless, $\mu_{0}H_{c2}\left(0\right)=12.2\left(1\right)$ T is still much lower than the upper critical fields observed for conventional superconductors such as NbTi (14.4 T) and Nb${}_{3}$Sn (28 T) [42].

The Maki parameter provides important insights into the mechanism of Cooper pair breakup in a magnetic field. It can be defined as [41]:(4)$$\alpha_{M}=\sqrt{2}\frac{H_{c2}^{\mathrm{orb}}}{H_{P}},$$
where $H_{c2}^{\mathrm{orb}}$ is the orbital-limited upper critical field and $H_{P}$ is the Pauli limiting field. Because $\alpha_{M}=1.4$ for Nb${}_{0.34}$Ti${}_{0.33}$Zr${}_{0.14}$Ta${}_{0.11}$Hf${}_{0.08}$, it is clear that $H_{c2}^{\mathrm{orb}}\approx H_{P}$. The temperature dependence of $\mu_{0}H_{c2}^{\mathrm{orb}}$ derived from the WHH model assuming the absence of Pauli spin paramagnetism ($\alpha_{M}=0$) and spin-orbit scattering ($\lambda_{\mathrm{SO}}=0$) is shown in Figure 7. The calculated value of $\mu_{0}H_{c2}^{\mathrm{orb}}\left(0\right)=13.3$ T is comparable to the Pauli limiting field $\mu_{0}H_{P}=13.4\left(2\right)$ T, estimated by the relation [43]:(5)$$\mu_{0}H_{P}=1.84T_{c}$$
or $\mu_{0}H_{P}=14.6\left(1\right)$ T, derived from the formula:(6)$$\mu_{0}H_{P}=\frac{\Delta_{0}}{\sqrt{2}\mu_{B}},$$
where $\mu_{B}$ is the Bohr magneton, and $\Delta_{0}$ is the superconducting energy gap calculated in the later part of this paper.

### 3.5. Thermodynamic Properties

The temperature dependence of the specific heat $C_{p}$ measured for Nb${}_{0.34}$Ti${}_{0.33}$Zr${}_{0.14}$Ta${}_{0.11}$Hf${}_{0.08}$ at zero magnetic field is shown in Figure 8. As can be seen, $C_{p}$(T) is featureless above $T_{c}$ and can be reproduced quite well using the free electron Sommerfeld model and the (oversimplified) Debye model for lattice vibrations (further corrected for anharmonicity [44]), represented by the first and second terms in the formula, respectively:(7)$$C_{p}\left(T\right)=\gamma T+\frac{9Rn}{1+\alpha T}{\left(\frac{T}{\Theta_{D}^{\mathrm{HT}}}\right)}^{3}\int_{0}^{\Theta_{D}^{\mathrm{HT}}/T}\frac{x^{4}e^{x}}{{\left(e^{x}-1\right)}^{2}}dx.$$

$\Theta_{D}^{\mathrm{HT}}$ is the Debye temperature (the superscript HT indicates that the parameter was estimated from high-temperature data), *n* is the number of atoms in the formula unit (f.u.), *R* denotes the universal gas constant, and α is the parameter describing the anharmonicity of the system. Taking n=1 for Nb${}_{0.34}$Ti${}_{0.33}$Zr${}_{0.14}$Ta${}_{0.11}$Hf${}_{0.08}$ and the Sommerfeld coefficient γ=6.96 mJ K${}^{-2}$ mol${}^{-1}$ (determined later in this section), the least squares fit of Equation (Equation 7) to the experimental data yields $\Theta_{D}^{\mathrm{HT}}=259\left(2\right)$ K and $\alpha =2.3\left(3\right)\times 10^{-4}$ K${}^{-1}$.

At low temperatures, the experimental data plotted as $C_{p}/T$ vs. $T^{2}$ (the bottom inset of Figure 8) show a sharp, discontinuous jump of $C_{p}$ at the SC transition temperature $T_{c}=7.5\left(2\right)$ K. First, one can consider the values of the normal state specific heat measured in the range between 7.7 and 10 K, which is described by the $T^{3}$-Debye law:(8)$$\frac{C_{p}\left(T\right)}{T}=\gamma +\beta T^{2},$$
with the Sommerfeld coefficient γ=6.96(12) mJ K${}^{-2}$ mol${}^{-1}$ and the lattice specific heat coefficient β=0.202(1) mJ K${}^{-4}$ mol${}^{-1}$. Using the β value, one can calculate the low temperature Debye temperature $\Theta_{D}^{\mathrm{LT}}=213\left(1\right)$ K from the expression:(9)$$\Theta_{D}^{\mathrm{LT}}={\left(\frac{12nR\pi^{4}}{5\beta }\right)}^{\frac{1}{3}}.$$

At the same time, γ can be used to compute the density of states (DOS) of the conduction electrons for both spin directions at the Fermi level $N(E_{F})$ by the relation:(10)$$\gamma =\frac{1}{3}\pi^{2}k_{B}^{2}N_{A}N\left(E_{F}\right),$$
where $k_{B}$ and $N_{A}$ are the Boltzmann constant and the Avogadro number, respectively. The determined value of γ corresponds to $N\left(E_{F}\right)=2.95\left(5\right)$ states/ eV per f.u.

The specific heat jump at $T_{c}$, $\Delta C_{p}\left(T_{c}\right)=82$ mJ mol${}^{-1}$ K${}^{-1}$, is estimated as the difference between the specific heats of the superconducting and normal states measured at T=7.2 K and T=7.7 K, respectively. The normalized jump $\Delta C_{p}\left(T_{c}\right)/\left(\gamma T_{c}\right)=1.57\left(5\right)$ is in reasonable agreement with the Bardeen–Cooper–Schrieffer (BCS) prediction, which assumes a value 1.43 [45], indicating that the studied HEA is close to the BCS type. Another important observation is that below $T_{c}$, the $C_{p}/T$ values tend to zero as $T^{2}$ decreases. This behavior demonstrates that there is no normal state electron contribution to the specific heat and that SC is a bulk effect, with the entire sample becoming SC below $T_{c}$ (the superconducting volume fraction is close to 100%).

The determined $T_{c}$ and $\Theta_{D}^{\mathrm{LT}}$ values are used to calculate the electron–phonon coupling constant $\lambda_{\mathrm{el}-\mathrm{ph}}$ using the McMillan’s formula [46]:(11)$$\lambda_{\mathrm{el}-\mathrm{ph}}=\frac{1.04+\mu^{*}\ln\left(\frac{\Theta_{D}^{\mathrm{LT}}}{1.45T_{c}}\right)}{\left(1-0.62\mu^{*}\right)\ln\left(\frac{\Theta_{D}^{\mathrm{LT}}}{1.45T_{c}}\right)-1.04},$$
where $\mu^{*}$ is the Coulomb pseudopotential, typically set between 0.1 and 0.15. Taking $\mu^{*}=0.125$, the calculated $\lambda_{\mathrm{el}-\mathrm{ph}}=0.71\left(1\right)$ indicates an intermediate electron–phonon coupling superconductivity in Nb${}_{0.34}$Ti${}_{0.33}$Zr${}_{0.14}$Ta${}_{0.11}$Hf${}_{0.08}$. This value is comparable to those reported for the other type-A HEA superconductors [7,12].

In the next step, by subtracting the phonon contribution βT from the total specific heat, one can calculate the temperature dependence of the superconducting state electron contribution $C_{\mathrm{es}}$ to the specific heat. The calculated experimental $C_{\mathrm{es}}\left(T\right)$ dependence is plotted as $\ln(C_{\mathrm{es}}/\gamma T_{c})$ vs. $T^{-1}$ in the top inset of Figure 8. As can be seen, the experimental data are well reproduced by the BCS description of the electronic specific heat [45]:(12)$$C_{\mathrm{BCS}}\left(T\right)=A\gamma T_{c}\exp\left(-\frac{\Delta_{0}}{k_{B}T}\right),$$
where *A* is a constant and $\Delta_{0}$ is the superconducting energy gap. From this analysis, it is found that the normalized energy gap $2\Delta_{0}/k_{B}T_{c}=3.7\left(1\right)$ is again in good agreement with the value of 3.52 predicted by the BCS theory for a single and isotropic gap in the weak coupling limit [45].

The thermodynamic critical field at 0 K $\mu_{0}H_{c}\left(0\right)$ can be estimated by considering the relation [47]:(13)$$\mu_{0}H_{c}\left(0\right)=\sqrt{\frac{3\gamma_{V}}{2\pi^{2}\mu_{0}}}\frac{\Delta_{0}}{k_{B}}.$$

Using the Sommerfeld coefficient per volume, $\gamma_{V}=605\left(10\right)$ J m${}^{-3}$ K${}^{2}$, and $\Delta_{0}/k_{B}=13.9\left(1\right)$ K, we find that $\mu_{0}H_{c}\left(0\right)=0.149\left(3\right)$ T.

From the obtained values of $H_{c}\left(0\right)$, $H_{c2}\left(0\right)$ and $H_{c2}^{\mathrm{orb}}$ one can also estimate the Ginzburg–Landau coherence length $\xi_{\mathrm{GL}}$ [48]:(14)$$\begin{array}{c} \xi_{\mathrm{GL}}=\sqrt{\frac{\phi_{0}}{2\pi \mu_{0}H_{c2}\left(0\right)}} \end{array}$$
and the Ginzburg–Landau penetration depth $\lambda_{\mathrm{GL}}$:(15)$$\begin{array}{c} \lambda_{\mathrm{GL}}=\sqrt{\frac{\phi_{0}\mu_{0}H_{c2}^{\mathrm{orb}}\left(0\right)}{4\pi \mu_{0}H_{c}{\left(0\right)}^{2}}}, \end{array}$$
where $\phi_{0}$ is the magnetic flux quantum. The calculated values for Nb${}_{0.34}$Ti${}_{0.33}$Zr${}_{0.14}$Ta${}_{0.11}$Hf${}_{0.08}$ are $\xi_{\mathrm{GL}}=5.2\left(1\right)$ nm and $\lambda_{\mathrm{GL}}=314\left(3\right)$ nm. From these values, the Ginzburg–Landau parameter $\kappa_{\mathrm{GL}}$, defined as:(16)$$\kappa_{\mathrm{GL}}=\frac{\lambda_{\mathrm{GL}}}{\xi_{\mathrm{GL}}},$$
can be determined. The obtained $\kappa_{\mathrm{GL}}=60.5\left(8\right)$ indicates once again that the studied HEA is a type II superconductor. Finally, using the formula [49]:(17)$$H_{c1}\left(0\right)=H_{c}\left(0\right)\frac{\ln\kappa_{\mathrm{GL}}}{\sqrt{2}\kappa_{\mathrm{GL}}},$$
we can deduce that $H_{c1}\left(0\right)=7.1\left(1\right)$ mT, and this value is in good agreement with the magnetization data.

### 3.6. Electronic Structure Calculations

The total and partial atomic densities of states of the Nb${}_{0.34}$Ti${}_{0.33}$Zr${}_{0.14}$Ta${}_{0.11}$Hf${}_{0.08}$ alloy calculated by the KKR-CPA method are shown in Figure 9. The shape of the total density of states (TDOS) is quite different from those previously reported for other type-A HEA superconductors [12,50,51]. In particular, the calculated TDOS for Ta${}_{0.34}$Nb${}_{0.33}$Hf${}_{0.08}$Zr${}_{0.14}$Ti${}_{0.11}$ [50] and (TaNb)${}_{0.67}$(HfZrTi)${}_{0.33}$ [51] reveal the presence of two pronounced peaks in the energy region close to the Fermi level ($E-E_{F}$ from −2 to 2 eV). One of them is close to the Fermi level $E_{F}$, while the second, the more pronounced one, is observed about 1 eV below $E_{F}$. In the case of TDOS obtained for Nb${}_{0.34}$Ti${}_{0.33}$Zr${}_{0.14}$Ta${}_{0.11}$Hf${}_{0.08}$, the Fermi level lies 0.05 eV above the TDOS maximum. The largest contribution to TDOS comes from the Nb and Ti atoms due to their highest atomic concentrations in the alloy. However, it should be noted that Ti has the highest partial atomic density of states near $E_{F}$ since this element has the most pronounced DOS peak in this energy region. At the same time, the shapes of Ta, Hf, Zr, and Nb DOS are quite similar to each other. The density of states at $E_{F}$ calculated by the KKR-CPA method is 1.98 states eV${}^{-1}$ f.u.${}^{-1}$. Substituting this value into Equation (Equation 10), one obtains the theoretical specific heat coefficient $\gamma_{\mathrm{th}}^{\mathrm{KKR}}=4.67$ mJ K${}^{-2}$ mol${}^{-1}$. A comparison to the experimental value of γ=6.96(12) mJ K${}^{-2}$ mol${}^{-1}$ gives the electron–phonon coupling constant $\lambda_{\mathrm{el}-\mathrm{ph}}^{\mathrm{KKR}}=\gamma /\gamma_{\mathrm{th}}^{\mathrm{KKR}}-1=0.49$ [50,52]. Surprisingly, this value is much lower than the experimentally derived $\lambda_{\mathrm{el}-\mathrm{ph}}=0.71\left(1\right)$. Consequently, using McMillan’s formula (Equation (Equation 11)) with $\mu^{*}=0.125$ and $\Theta_{D}^{\mathrm{LT}}=213\left(1\right)$ K, the calculated $T_{c}$ is close to 1 K. Obviously, this value is extremely low compared to $T_{c}$ determined from experimental data. Since then, the KKR-CPA results have been additionally verified by supercell DFT calculations within the plane-wave function basis as implemented in the VASP.

Total and partial atomic densities of states of the Nb${}_{0.34}$Ti${}_{0.33}$Zr${}_{0.14}$Ta${}_{0.11}$Hf${}_{0.08}$ HEA calculated by the PAW method for three assumed random atomic arrangements are shown in the top panel of Figure 10. As can be seen, the DOS obtained for all three random atomic arrangements are very similar to each other, confirming our previous finding [12] that the local atomic configuration does not play a crucial role in determining the electronic structure of the HEA in the energy region close to $E_{F}$. The averaged TDOS is shown in the bottom panel of Figure 10 together with TDOS calculated by the KKR-CPA method. The relatively small difference between these two curves indicates that both methods give consistent results. The density of states at $E_{F}$ calculated from the VASP data is 2.12 states eV${}^{-1}$ f.u.${}^{-1}$. Furthermore, using the same procedure which was previously applied to the KKR-CPA results, we obtain $\gamma_{\mathrm{th}}^{\mathrm{PAW}}=5$ mJ K${}^{-2}$ f.u.${}^{-1}$, $\lambda_{\mathrm{el}-\mathrm{ph}}^{\mathrm{PAW}}=0.39$ and $T_{c}<1$ K.

In summary, the results of the theoretical calculations are in contradiction with the experimental data. In particular, the calculated values of $\lambda_{\mathrm{el}-\mathrm{ph}}$ and $T_{c}$ are much lower than those obtained from magnetic properties, specific heat, and resistivity measurements. In fact, similar problems with reproducing experimental $T_{c}$ using theoretical calculations have also been reported in the literature for Nb${}_{3}$Ge [53], V [54] and MgCNi${}_{3}$ [55] as well as for several HEA superconductors [12,50,51]. However, it should be noted that previous KKR-CPA calculations performed for HEA superconductors gave overestimated values of $\lambda_{\mathrm{el}-\mathrm{ph}}$ and $T_{c}$ [12,50,51], while in this case both parameters are strongly underestimated. The explanation for the discrepancies obtained in this work is unknown, but they may be related to the “cocktail effect” of HEA [3,4] or to the presence of large amounts of crystallographic defects (vacancies, dislocations) in the studied sample. The second explanation is based on the fact that the electronic structure calculations assume the absence of any defects, while XRD data reveal a relatively high level of internal strains. At the same time, the influence of crystallographic defects on the superconducting properties of various superconductors has recently been reported in work [56,57,58].

## 4. Conclusions

The formation and physical properties of a novel high-entropy alloy Nb${}_{0.34}$Ti${}_{0.33}$Zr${}_{0.14}$Ta${}_{0.11}$Hf${}_{0.08}$ are presented and described. As expected for type-A HEA superconductors, the alloy crystallizes in a simple bcc structure. At the same time, a compositional analysis by EDXS shows a homogeneous distribution of alloying elements in the studied sample. The determined physical properties indicate that the alloy exhibits type II superconductivity with a conventional type II superconductor with a critical temperature $T_{c}$ close to 7.5 K, an intermediate electron–phonon coupling and an upper critical field $\mu_{0}H_{c2}=\begin{array}{c} 12.2(1) \end{array}$ T. This significant finding indicates that Nb${}_{0.34}$Ti${}_{0.33}$Zr${}_{0.14}$Ta${}_{0.11}$Hf${}_{0.08}$ has one of the highest $\mu_{0}H_{c2}$ in the family of type-A HEA superconductors.

## Figures and Tables

**Figure 1 materials-16-05814-f001:**
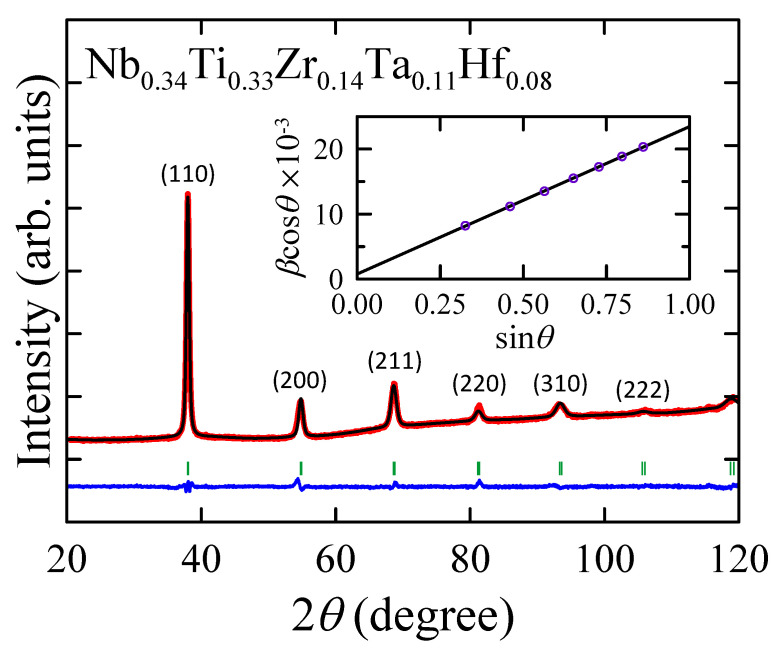
X-ray diffraction pattern collected at room temperature for Nb${}_{0.34}$Ti${}_{0.33}$Zr${}_{0.14}$Ta${}_{0.11}$Hf${}_{0.08}$. Red dots and black lines represent experimental data, and the result of the Rietveld refinement, respectively, and blue line shows the difference between the two. Green dashes indicate positions of the Bragg reflections. Inset shows the Williamson–Hall plot; solid line is a linear fit to data points.

**Figure 2 materials-16-05814-f002:**
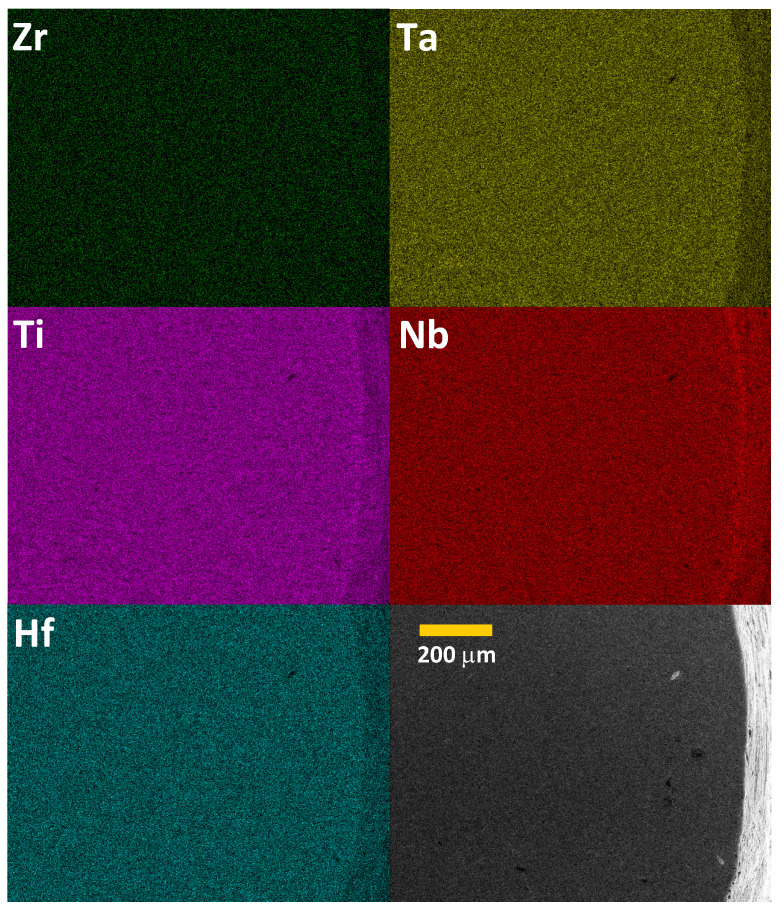
SEM micrograph (lower right image) and EDXS elemental mapping (other images) of Nb${}_{0.34}$Ti${}_{0.33}$Zr${}_{0.14}$Ta${}_{0.11}$Hf${}_{0.08}$.

**Figure 3 materials-16-05814-f003:**
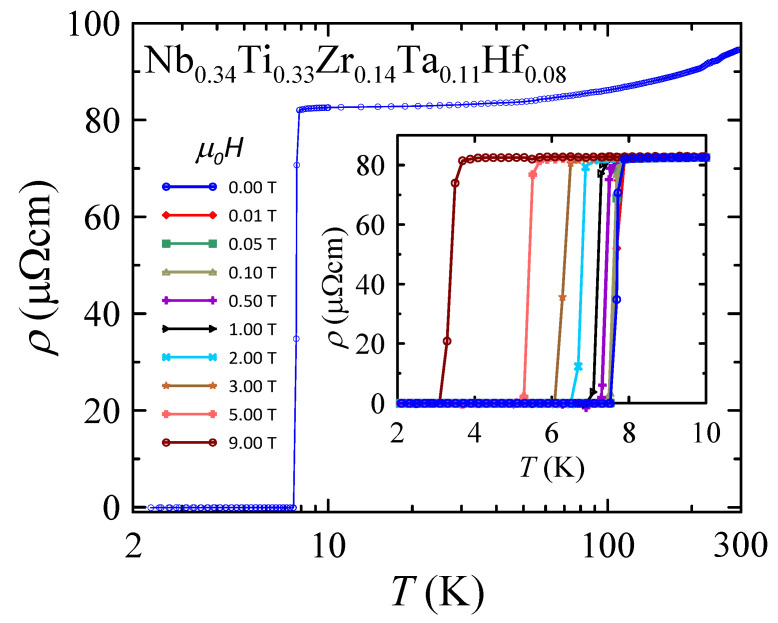
Temperature dependence of electrical resistivity in zero magnetic field measured for Nb${}_{0.34}$Ti${}_{0.33}$Zr${}_{0.14}$Ta${}_{0.11}$Hf${}_{0.08}$. The inset shows the resistivities in nominal applied magnetic fields $\mu_{0}H$ in temperature range of 2–10 K. Solid curves serve as guides for the eye.

**Figure 4 materials-16-05814-f004:**
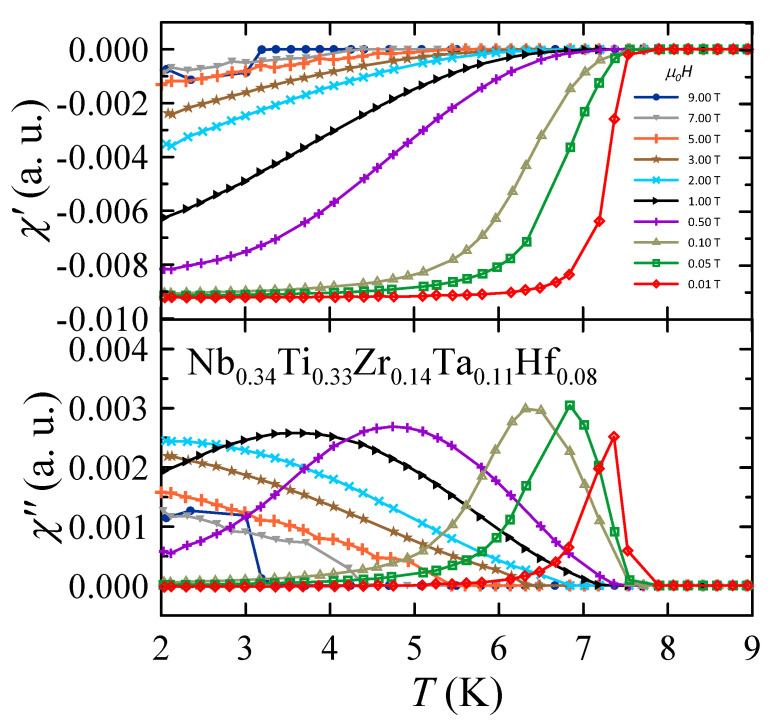
Real $\chi^{\prime }$ and imaginary $\chi^{\prime \prime }$ parts of AC susceptibility measured for the Nb${}_{0.34}$Ti${}_{0.33}$Zr${}_{0.14}$Ta${}_{0.11}$ Hf${}_{0.08}$ in nominal applied magnetic fields $\mu_{0}H$. Solid curves serve as guides for the eye.

**Figure 5 materials-16-05814-f005:**
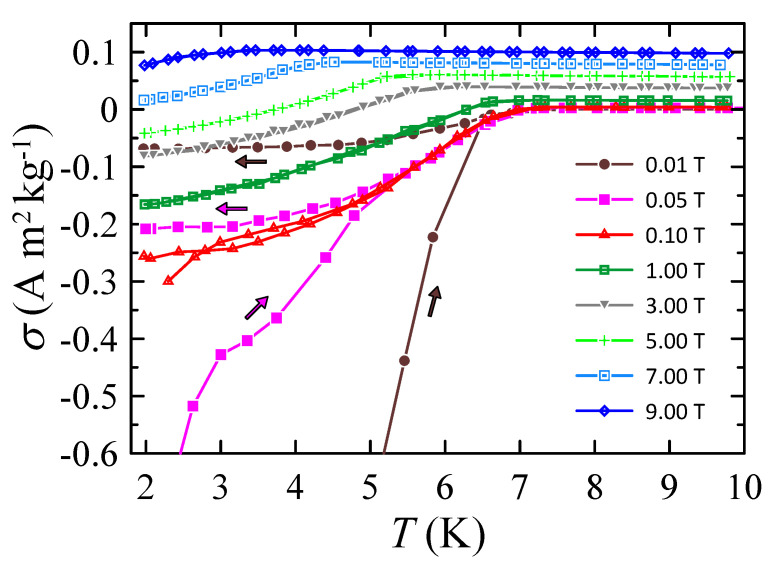
Temperature dependence of DC mass magnetization measured for Nb${}_{0.34}$Ti${}_{0.33}$Zr${}_{0.14}$Ta${}_{0.11}$ Hf${}_{0.08}$ in nominal applied magnetic fields $\mu_{0}H$. The arrows indicate the direction of temperature change. Solid curves serve as guides for the eye.

**Figure 6 materials-16-05814-f006:**
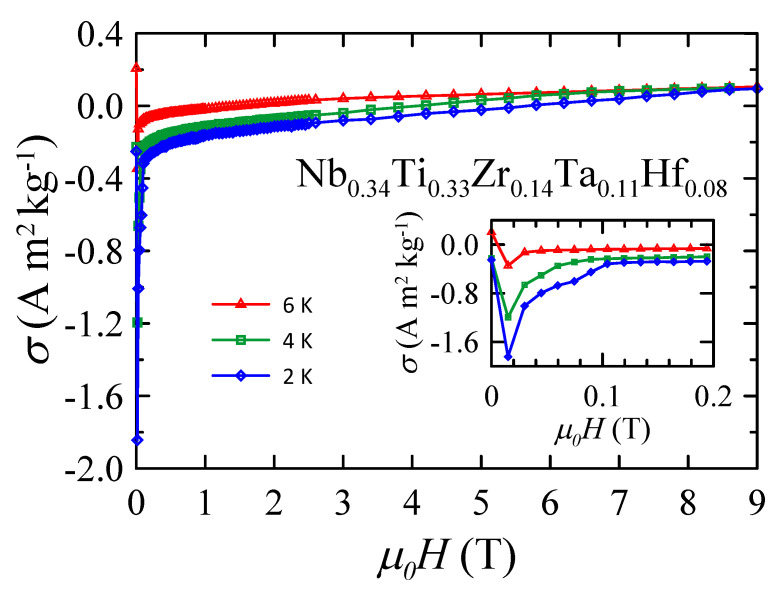
Magnetic field dependence of DC mass magnetization measured for Nb${}_{0.34}$Ti${}_{0.33}$Zr${}_{0.14}$Ta${}_{0.11}$ Hf${}_{0.08}$ at various temperatures. The inset shows the low field region. Solid curves serve as guides for the eye.

**Figure 7 materials-16-05814-f007:**
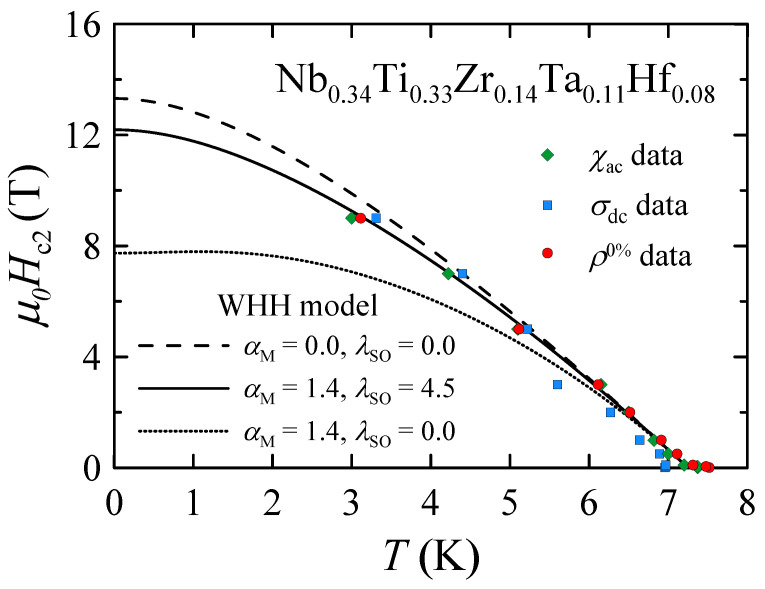
Temperature dependence of the upper critical fields $\mu_{0}H_{c2}$ for Nb${}_{0.34}$Ti${}_{0.33}$Zr${}_{0.14}$Ta${}_{0.11}$Hf${}_{0.08}$ derived from its AC susceptibility ($\chi_{\mathrm{AC}}$), DC mass magnetization ($\sigma_{\mathrm{DC}}$) and resistivity ($\rho^{0\%}$) measurements. Solid and dashed lines are fits of the WHH model (Equation (Equation 3)) to the experimental points.

**Figure 8 materials-16-05814-f008:**
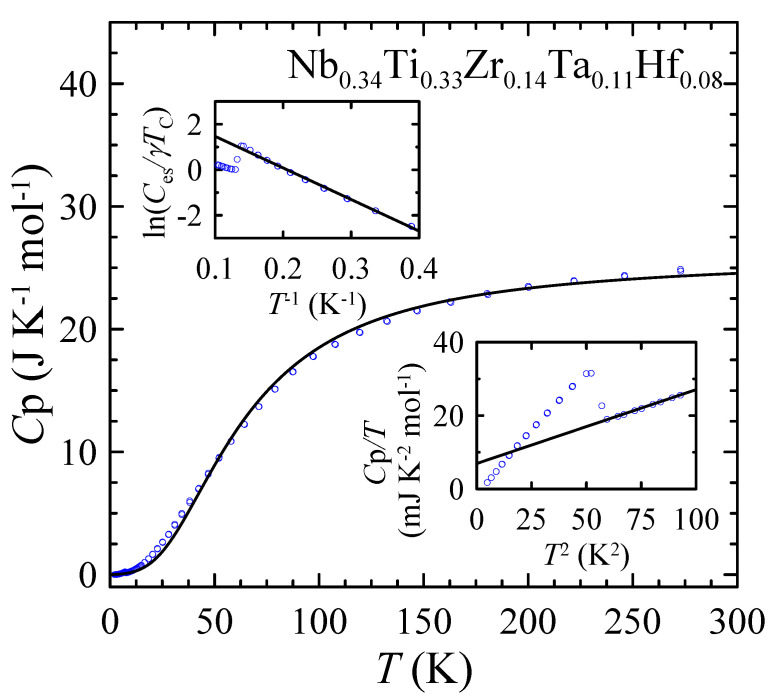
Temperature dependence of the specific heat $C_{p}$ measured for Nb${}_{0.34}$Ti${}_{0.33}$Zr${}_{0.14}$Ta${}_{0.11}$Hf${}_{0.08}$ at zero magnetic field. Solid line is fit of Equation (Equation 7) to the experimental points. Bottom inset shows low temperature specific heat data plotted as $C_{p}/T$ vs. $T^{2}$. Solid line is a linear fit of Equation (Equation 8) to the experimental points. Top inset shows $\ln(C_{\mathrm{es}}/\gamma T_{c})$ vs. $T^{-1}$. Solid line is a linear fit of Equation (Equation 12) to the experimental points.

**Figure 9 materials-16-05814-f009:**
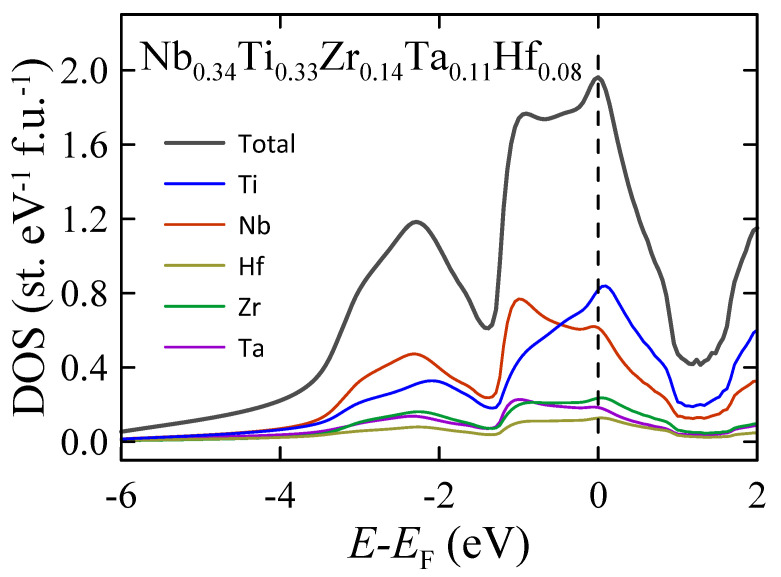
Electron density of states of Nb${}_{0.34}$Ti${}_{0.33}$Zr${}_{0.14}$Ta${}_{0.11}$Hf${}_{0.08}$ HEA calculated by the KKR-CPA method. Total DOS, plotted with gray solid line, and partial atomic densities, marked with colors and weighted by their atomic concentrations.

**Figure 10 materials-16-05814-f010:**
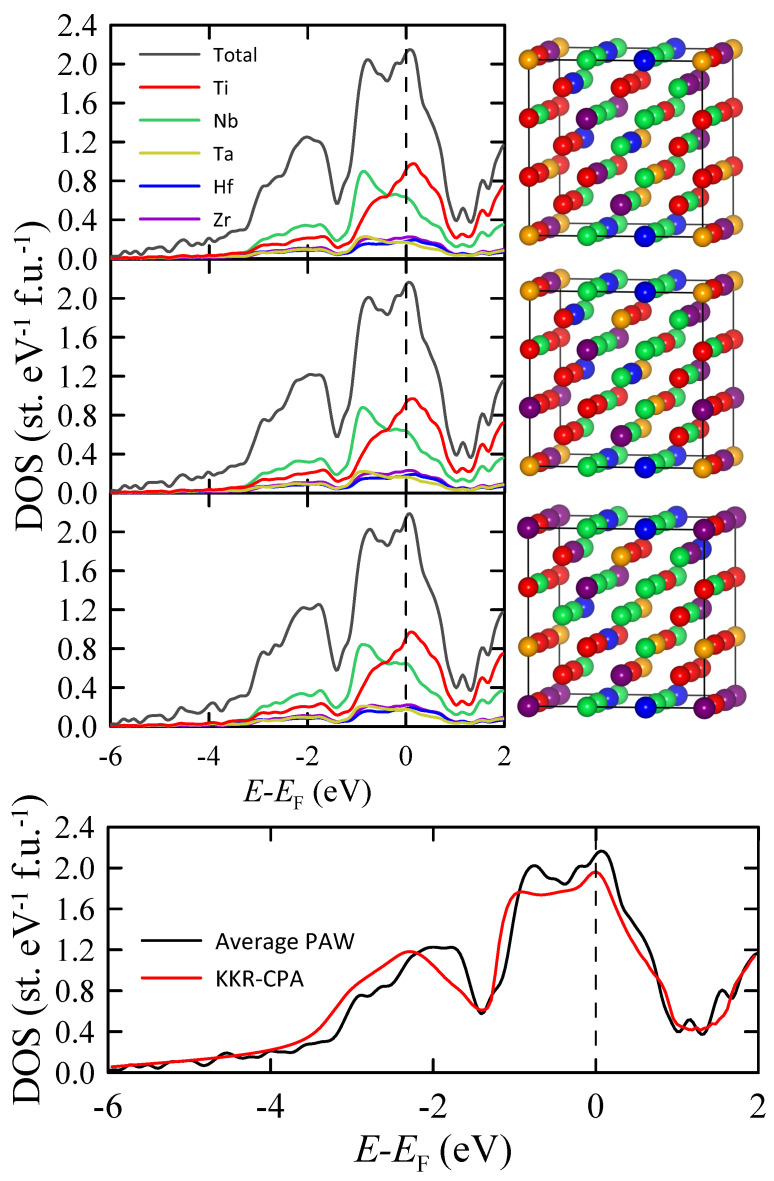
(**Top panel**) Total and partial atomic DOS of Nb${}_{0.34}$Ti${}_{0.33}$Zr${}_{0.14}$Ta${}_{0.11}$Hf${}_{0.08}$ HEA calculated by the PAW method using VASP for three assumed random atomic arrangements which are shown on the right. (**Bottom panel**) Comparison of TDOS averaged over three configurations with the KKR-CPA data.

## Data Availability

The data presented in this study are openly available in OSF repository at DOI 10.17605/OSF.IO/G4N6B.

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
