# Peer review of "Enhanced Superconducting Critical Parameters in a New High-Entropy Alloy Nb0.34Ti0.33Zr0.14Ta0.11Hf0.08"

_materials, 2023, doi:10.3390/ma16175814_

Round 1

Reviewer 1 Report

General remarks:
- A lot of papers on Ta-Nb-Hf-Zr-Ti HEAs have been published, so no real novelty in picking the general subject.
- What is good: It is true that this manuscript seems to be the only one intentionally aiming for compositions that are Ti rich in the Ta-Nb-Hf-Zr-Ti system. Even if not necessarily the absolute highest in type-A HEA superconductors, the resulting upper critical field of 13.5 T is 15 % or more better than what is obtained in Ta-Nb-Hf-Zr-Ti, so the idea has paid off.
- What is good: Nice to see an article contain at the same time experiments on superconductivity of HEAs plus theoretical calculations. We don't have many of those.
- Interesting: While the two theoretical calculations give consistent results on the electron-phonon constant, they do not match the experimental data. They're not even close.
- The bad: The discussion of the results seems to be quite confused in some parts and some shortcuts seem to have been cut when writing the manuscript.

I would propose that the article can be accepted if corrections are made and the quality of the manuscript is somewhat improved.

In this journal "minor revisions" implies accepting the article without another round of peer-review. I think that another round is necessary in this case. However, I do imagine that the second round of review would likely be very trivial.

-------------------------------------------------------

Things that need to be fixed or checked:

1.) A general remark: You're designating the alloy as Ti0.33Nb0.34Hf0.08Zr0.14Ta0.11. The order of the elements seems puzzling to me. Could you elaborate on the order, namely why Ti-Nb-Hf-Zr-Ti and not e.g. Ta-Nb-Hf-Zr-Ti or Ta-Nb-Zr-Hf-Ti as previously used in literature?

2.) Perhaps references to
Kim et al (https://doi.org/10.1016/j.actamat.2020.01.007) and Yuan et al (doi: 10.3389/fmats.2018.00072) might be relevant in the introduction (around page 1, line 25). Both list Hc2's of about 12 T.

3.) page 2/line 37 "superconductivity in Ti0.33Nb0.34Hf0.08Zr0.14Ta0.11 below 7 K"
Your transitions are at 7.4 K or 7.8 K. Shouldn't this say "at about 7.5 K" or "below 7.5 K" or something similar?

4.) page 2/lines 45-47. What form were the elements that you used for synthesis? Powders, sheet, wire, slugs?

5.) page 2/lines 58-60: More data is needed, so that readers can figure out how exactly you did your experiments about physical properties. Some of the missing data also has bearing for the interpretation of the results you provide.
A minimum that we need is:
    - Sample shape and direction for resistivity measurements - e.g. a long bar of dimensions 8 mm x 1 mm x 1 mm. The bar was oriented with the long side perpendicular/parallel to the magnetic field. Please also mention that you were using a four terminal/contact method.
    - What was the piece for magnetic measurements like? Was is regularly shaped or irregular, long direction along or perpendicular to the field? Was is a sphere, a long rod? What was the demagnetizing factor, did your take it into account?
   - How you selected pieces for the different measurements and for the material characterization? Were the pieces for magnetization, specific heat, electrical resistivity the same piece, neighboring pieces or pieces from completely different parts of the sample?

6.) page 2/around line 80: You never mention whether you relaxed the atomic positions or not. What I am trying to say is: Did you optimize just the size of the unit cell or did you also optimize the position of each atom inside the cell so that the force on it is zero (meaning it is at its equilibrium position)?

7.) page 2/lines 82-84 "The lattice parameters derived using the third-order Birch-Murnaghan equation of state were for all three configurations close to aPAW = 3.34 Å  and BPAW = 122.5 GPa." You probably did this  as a check to show that the KKR and DFT give very similar unit cells sizes and bulk modulii, meaning that they are compatible. If so, you could mention this.

8.) page 3/figure 1. Not a fan of the size of the red circles - it seems like they are hiding the data quality. Either make them smaller or us a thick full line

9.) section 3.2. I am not entirely convinced of sample homogeneity if just shown a SEM micrograph over a whole milimeter. I assume that you have done also images at higher magnification, e.g. ones where the bar showing the scale would be 1 micrometer or 10 micrometers? Could you include those at least in the supplementary material and/or add a very brief comment on them in the main text? Things like lamellar decomposition and eutectic decomposition pop-up in high-entropy alloys and it would be information that would be good to have.

10.) page 5/figure 3. The 0T line in the main panel seems to have a double line at the transition. You might have forgotten to remove a point when merging datasets.

11.) page 6/figure 4. Would be better to switch the panels - the real part should be the upper panel and the imaginary part the lower one. This is the way you mention it in the text and the caption of the image. Also "0,50 T" and "0,05 T" should contain a dot not a comma.

12.) page 6/lines 150-182 "These values are slightly lower than Tc obtained from electrical resistivity measurements. Therefore, this finding suggests some contribution of the surface superconductivity in the studied HEA which often is observed in bulk superconductors".
This I'm not sure that I can agree with. If I take your imaginary part chi'', I see that it starts going up already at 7.8 K for 0.01 T, which I can interpret as Cooper pairs forming. Meaning that I have explained away all the difference you see. And this is before I start discussing questions of sample homogeneity (especially if you used different pieces for your magnetism and electrical resistivity) and residual fields in your magnet (can also be on the order of 40 oersted). If you just look at the width of the peaks in your XRD, perfect sample homogeneity is probably out of the question.

13.) page 6/154 and onwards. Magnetization in recent articles is typically set as M and not sigma. Either fix or elaborate where this notation comes from.

14.) page 8/lines 162-163: "The data obtained at low-field region show that Ti0.33Nb0.34Hf0.08Zr0.14Ta0.11 162 is the type-II superconductor with the lower critical field μ0Hc1 ≈ 15 mT at 2 K." If possible, you should repeat the measurements in the inset of your Fig 6. I'm not talking about collecting the whole hysteresis loops, just the part to 0.2 T with more data. Currently we can only read mu0Hc1 as  15 mT +/- 15 mT as we have points only spaced at 15 mT.

15.) pages 6 to 8: I don't see in any place a DC susceptility mentioned. Is it -1 corresponding to a full Meissner effect at 15mT or much different? This is really something we should have in the article.

16.) Page 8/Fig 7 and Page 8 lines 175 - 183: The designations of Hc2 and Hc3 are not very good as far as I am concerned. If you were discussing surface superconductivity to a greater extent and you did more experiments (and more detailed), you might be able to actually determine Hc3 and plotting that one separately. Looking at you data, I'd actually suggest calling one Hc2^mag and the other Hc2^rho. In reality, both experiments probably just determined Hc2.

17.) Page 8/lines 182 to Page 9/line 192. I think this section needs some checking or some additional explanation. The critical field that you determine for your sample is 13.5 T. You calculate the orbital limiting field as 9.7 T and call this close. I'd say you either need a better model (to get a higher orbital limiting field) or a good comment explaining what you think is happening.
Also: I don't see anywhere clearly stating what you think is the dominant pair-breaking mechanism. You only mention a moderate Pauli limiting effect.

18.) Page 9/lines 195-204 The Debye fit in Fig8 doesn't look too good to be honest - there is a lot of deviation - so it doesn't describe the Cp data well. Secondly, I don't find a place where you would actually use the data you obtain from the fit. Please add some interpretation, move to supplementary or remove altogether.

19.) Page 10/Fig 8. You have a weird number of minor ticks on both axes. On the horizontal you are going 12.5 K, 25 K, 37.5 K, 50 K, etc. On the vertical 2.5, 5, 7.5, 10.

20.) Page 10/lines 216-218 I don't see a point in calculating the Pauli susceptibility if you don't provide any susceptibility data that I could interpret using it.

21.) Page 12/262 "the Fermi level lies 0.05 eV above the TDOS maximum, which is usually favorable for superconductivity"
Citation missing?

22.) Page 14/289-298
The discrepancy for the electron-phonon coupling constant is surprisingly large.
- You have re-checked the conversions (specific heat per mass to specific heat per mole, etc.) for numbers for gamma both from experiment and the theoretical calculation, correct?
- Unusual in your case seems to be that you underestimate the transition temperature Tc when you try to calculate if from theorical calculations (and using McMillan's formula). All the other references that try to do this - both articles by Jasiewicz et al. (your references 44,45) and your own work (Sobota, ref. 31) - get exactly the opposite, an overestimated transition temperature. This should be explicitly mentioned as it might be important in the future - potentially at some point it might help us understand why ab initio calculations for HEAs work only to some accuracy and how this can be improved.

23.) Page 14/289-298
(This is connected to a previous comment that I had made in the Methods section.)
One thing both your methods (KKR-CPA and DFT with PAW) seem to have in common is that the atoms are at their ideal positions. It would be worth relaxing the atoms in the DFT to their equilibrium positions (the positions where the forces on them are zero) and checking if this makes your theoretical density of states more similar to the experimental one.

24.) Page 14/302 "while a microprobe analysis shows" You probably meant composition analysis by SEM EDX or similar.

25.) Page 14/306
"This significant finding indicates that Ti0.33Nb0.34Hf0.08Zr0.14Ta0.11 exhibits the highest μ0Hc2 in the family of type-A HEA superconductors." What about Krnel et al. https://doi.org/10.3390/ma15031122? Probably you just need to rephrase slightly.

Author Response

Letter of response

We thank very much referee for carefully reading the paper and for numerous comments. In all cases they were taken into account.

General remarks:

- A lot of papers on Ta-Nb-Hf-Zr-Ti HEAs have been published, so no real novelty in picking the general subject.

- What is good: It is true that this manuscript seems to be the only one intentionally aiming for compositions that are Ti rich in the Ta-Nb-Hf-Zr-Ti system. Even if not necessarily the absolute highest in type-A HEA superconductors, the resulting upper critical field of 13.5 T is 15 % or more better than what is obtained in Ta-Nb-Hf-Zr-Ti, so the idea has paid off.

- What is good: Nice to see an article contain at the same time experiments on superconductivity of HEAs plus theoretical calculations. We don't have many of those.

- Interesting: While the two theoretical calculations give consistent results on the electron-phonon constant, they do not match the experimental data. They're not even close.

- The bad: The discussion of the results seems to be quite confused in some parts and some shortcuts seem to have been cut when writing the manuscript.

I would propose that the article can be accepted if corrections are made and the quality of the manuscript is somewhat improved.

In this journal "minor revisions" implies accepting the article without another round of peer-review. I think that another round is necessary in this case. However, I do imagine that the second round of review would likely be very trivial.

-------------------------------------------------------

Things that need to be fixed or checked:

1.) A general remark: You're designating the alloy as Ti0.33Nb0.34Hf0.08Zr0.14Ta0.11. The order of the elements seems puzzling to me. Could you elaborate on the order, namely why Ti-Nb-Hf-Zr-Ti and not e.g. Ta-Nb-Hf-Zr-Ti or Ta-Nb-Zr-Hf-Ti as previously used in literature?

Discussion: Indeed, we changed the order of the elements. We chose Nb0.34Ti0.33Zr0.14Ta0.11 Hf0.08 (the first with the highest concentration and the last with the lowest one). 

2.) Perhaps references to

Kim et al (https://doi.org/10.1016/j.actamat.2020.01.007) and Yuan et al (doi: 10.3389/fmats.2018.00072) might be relevant in the introduction (around page 1, line 25). Both list Hc2's of about 12 T.

Discussion: Both articles have been added to the introduction.

3.) page 2/line 37 "superconductivity in Ti0.33Nb0.34Hf0.08Zr0.14Ta0.11 below 7 K"

Your transitions are at 7.4 K or 7.8 K. Shouldn't this say "at about 7.5 K" or "below 7.5 K" or something similar?

Discussion: Corrected.

4.) page 2/lines 45-47. What form were the elements that you used for synthesis? Powders, sheet, wire, slugs?

Discussion: We added that the alloy was prepared by conventional arc melting method from high-purity metals in the form of slugs and chips.

5.) page 2/lines 58-60: More data is needed, so that readers can figure out how exactly you did your experiments about physical properties. Some of the missing data also has bearing for the interpretation of the results you provide.

A minimum that we need is:

    - Sample shape and direction for resistivity measurements - e.g. a long bar of dimensions 8 mm x 1 mm x 1 mm. The bar was oriented with the long side perpendicular/parallel to the magnetic field. Please also mention that you were using a four terminal/contact method.

    - What was the piece for magnetic measurements like? Was is regularly shaped or irregular, long direction along or perpendicular to the field? Was is a sphere, a long rod? What was the demagnetizing factor, did your take it into account?

   - How you selected pieces for the different measurements and for the material characterization? Were the pieces for magnetization, specific heat, electrical resistivity the same piece, neighboring pieces or pieces from completely different parts of the sample?

Discussion: All necessary data about measurements of physical properties of the studied sample have been added in the revised manuscript.

6.) page 2/around line 80: You never mention whether you relaxed the atomic positions or not. What I am trying to say is: Did you optimize just the size of the unit cell or did you also optimize the position of each atom inside the cell so that the force on it is zero (meaning it is at its equilibrium position)?

Discussion: In both KKR-CPA and DFT atoms are assumed to occupy high-symmetry positions of the bcc lattice. Their positions were not optimized and only the size of the unit cell was optimized.  In the case of the KKR method it is not possible to optimize the atomic position as the unit cell contains only one “atom” made of constituent elements in the proportion corresponding to the stoichiometry of the HEA. In the case of DFT calculations, the atomic optimization would have to be made of each atomic configuration separately, which would significantly increase the computational cost. Moreover, we think it is reasonable to expect that displacements for ideal bcc positions would average out once many atomic configurations were considered.

Please see also answer 23) where this issue was further addressed.

7.) page 2/lines 82-84 "The lattice parameters derived using the third-order Birch-Murnaghan equation of state were for all three configurations close to aPAW = 3.34 Å  and BPAW = 122.5 GPa." You probably did this  as a check to show that the KKR and DFT give very similar unit cells sizes and bulk modulii, meaning that they are compatible. If so, you could mention this.

Discussion: It is added that the parameters derived using the third-order Birch-Murnaghan equation of state obtained from PAW are in good agreement with corresponding parameters obtained from KKR-CPA.

8.) page 3/figure 1. Not a fan of the size of the red circles - it seems like they are hiding the data quality. Either make them smaller or us a thick full line

Discussion: We prepared a new Fig. 1 with smaller red circles.

9.) section 3.2. I am not entirely convinced of sample homogeneity if just shown a SEM micrograph over a whole milimeter. I assume that you have done also images at higher magnification, e.g. ones where the bar showing the scale would be 1 micrometer or 10 micrometers? Could you include those at least in the supplementary material and/or add a very brief comment on them in the main text? Things like lamellar decomposition and eutectic decomposition pop-up in high-entropy alloys and it would be information that would be good to have.

Discussion: We have added supplementary materials in which we present all EDS elemental maps of the investigated HEA. The obtained results show the same concentration of elements. Therefore, we believe that our alloy is homogenous on the micrometer scale. XRD data support this scenario. There is a possibility for slight phase segregation on the nanometer scale, as observed in other HEA superconductors. At the same time, we observed quite sharp superconducting transitions at 0 T which suggest that this HEA is likely to be homogeneous even on the nanometer scale as well.

10.) page 5/figure 3. The 0T line in the main panel seems to have a double line at the transition. You might have forgotten to remove a point when merging datasets.

Discussion: Corrected.

11.) page 6/figure 4. Would be better to switch the panels - the real part should be the upper panel and the imaginary part the lower one. This is the way you mention it in the text and the caption of the image. Also "0,50 T" and "0,05 T" should contain a dot not a comma.

Discussion: Corrected.

12.) page 6/lines 150-182 "These values are slightly lower than Tc obtained from electrical resistivity measurements. Therefore, this finding suggests some contribution of the surface superconductivity in the studied HEA which often is observed in bulk superconductors".

This I'm not sure that I can agree with. If I take your imaginary part chi'', I see that it starts going up already at 7.8 K for 0.01 T, which I can interpret as Cooper pairs forming. Meaning that I have explained away all the difference you see. And this is before I start discussing questions of sample homogeneity (especially if you used different pieces for your magnetism and electrical resistivity) and residual fields in your magnet (can also be on the order of 40 oersted). If you just look at the width of the peaks in your XRD, perfect sample homogeneity is probably out of the question.

Discussion: We disagree with that statement. In fact, the measured values of chi’ and chi’’ above 8 K at 0.01 T are close to 1,61(5)*10^-6 and 6,91(5)*10^-6, respectively. At 7.89 K and 0.01 T chi’ and chi’’ are equal to 1,61*10^-6 and 6,93*10^-6. In our opinion these values are similar to each other and do not indicate the formation of Cooper pairs. As we mentioned above, for magnetic and resistivity measurements we used the same sample. The width of the peaks in our XRD is mainly connected with a non-uniform lattice strain which is rather obvious in the case of as-prepared (without annealing) HEA (chemical disorder, lattice defects).

13.) page 6/154 and onwards. Magnetization in recent articles is typically set as M and not sigma. Either fix or elaborate where this notation comes from.

Discussion: We agree that magnetization is typically set as M. However, in our paper we present values of mass magnetization/specific magnetization which is typically set as sigma. This information has been added in the revised manuscript.

14.) page 8/lines 162-163: "The data obtained at low-field region show that Ti0.33Nb0.34Hf0.08Zr0.14Ta0.11 162 is the type-II superconductor with the lower critical field μ0Hc1 ≈ 15 mT at 2 K." If possible, you should repeat the measurements in the inset of your Fig 6. I'm not talking about collecting the whole hysteresis loops, just the part to 0.2 T with more data. Currently we can only read mu0Hc1 as  15 mT +/- 15 mT as we have points only spaced at 15 mT.

Discussion: Unfortunately, we will not be able to repeat the experiments in the coming weeks. In any case, such measurements would not yield much new relevant information, since the measured magnetization signal could be affected by a different residual magnetic field or a slight tilt of the sample during its assembly for a new experiment. Moreover, the value of Hc1 = 7.1(1) mT is estimated using Orlando’s formula.

15.) pages 6 to 8: I don't see in any place a DC susceptility mentioned. Is it -1 corresponding to a full Meissner effect at 15mT or much different? This is really something we should have in the article.

Discussion: We added that taking the demagnetization factor N = 0.58, one can estimate the dimensionless volume susceptibility which is close to -0.73 in magnetic field of 15 mT at 2 K. This value is higher than the expected value of -1 for the fully developed Meissner state. Therefore, it is plausible to assume that the lower critical field Hc1 < 15 mT at 2 K.

16.) Page 8/Fig 7 and Page 8 lines 175 - 183: The designations of Hc2 and Hc3 are not very good as far as I am concerned. If you were discussing surface superconductivity to a greater extent and you did more experiments (and more detailed), you might be able to actually determine Hc3 and plotting that one separately. Looking at you data, I'd actually suggest calling one Hc2^mag and the other Hc2^rho. In reality, both experiments probably just determined Hc2.

17.) Page 8/lines 182 to Page 9/line 192. I think this section needs some checking or some additional explanation. The critical field that you determine for your sample is 13.5 T. You calculate the orbital limiting field as 9.7 T and call this close. I'd say you either need a better model (to get a higher orbital limiting field) or a good comment explaining what you think is happening.

Also: I don't see anywhere clearly stating what you think is the dominant pair-breaking mechanism. You only mention a moderate Pauli limiting effect.

Discussion – points 16-17: The section which describes magnetic results is significantly corrected. G-L formula is replaced by WHH model. The values of determined critical fields are corrected or/and recalculated.

18.) Page 9/lines 195-204 The Debye fit in Fig8 doesn't look too good to be honest - there is a lot of deviation - so it doesn't describe the Cp data well. Secondly, I don't find a place where you would actually use the data you obtain from the fit. Please add some interpretation, move to supplementary or remove altogether.

Discussion – We agree that the Debye fit doesn't describe the Cp data well. However we cannot propose any function/model which will better describe the measured Cp data. In our opinion this intriguing result is worth presenting to the readers.

19.) Page 10/Fig 8. You have a weird number of minor ticks on both axes. On the horizontal you are going 12.5 K, 25 K, 37.5 K, 50 K, etc. On the vertical 2.5, 5, 7.5, 10.

Discussion – On both axes we have the same number of minor ticks.

20.) Page 10/lines 216-218 I don't see a point in calculating the Pauli susceptibility if you don't provide any susceptibility data that I could interpret using it.

Discussion – Calculation of Pauli susceptibility is removed from the manuscript.

21.) Page 12/262 "the Fermi level lies 0.05 eV above the TDOS maximum, which is usually favorable for superconductivity"

Discussion – After discussion we decided to remove the second part of this sentence.

22.) Page 14/289-298

The discrepancy for the electron-phonon coupling constant is surprisingly large.

- You have re-checked the conversions (specific heat per mass to specific heat per mole, etc.) for numbers for gamma both from experiment and the theoretical calculation, correct?

Discussion – the calculations and conversions are correct.

- Unusual in your case seems to be that you underestimate the transition temperature Tc when you try to calculate if from theorical calculations (and using McMillan's formula). All the other references that try to do this - both articles by Jasiewicz et al. (your references 44,45) and your own work (Sobota, ref. 31) - get exactly the opposite, an overestimated transition temperature. This should be explicitly mentioned as it might be important in the future - potentially at some point it might help us understand why ab initio calculations for HEAs work only to some accuracy and how this can be improved.

Discussion – This fact is mentioned in the revised manuscript.

23.) Page 14/289-298

(This is connected to a previous comment that I had made in the Methods section.)

One thing both your methods (KKR-CPA and DFT with PAW) seem to have in common is that the atoms are at their ideal positions. It would be worth relaxing the atoms in the DFT to their equilibrium positions (the positions where the forces on them are zero) and checking if this makes your theoretical density of states more similar to the experimental one.

Discussion – Unfortunately, we were not able to run structure optimization for the DFT-PAW method due to the large computational cost and limited time for revision.

To assess the robustness of the DOS to the atomic displacement we did the experiment using the KKR-CPA framework. We have added supplementary materials in which we present this experiment and obtained results.

24.) Page 14/302 "while a microprobe analysis shows" You probably meant composition analysis by SEM EDX or similar.

Discussion – Corrected

25.) Page 14/306

"This significant finding indicates that Ti0.33Nb0.34Hf0.08Zr0.14Ta0.11 exhibits the highest μ0Hc2 in the family of type-A HEA superconductors." What about Krnel et al. https://doi.org/10.3390/ma15031122? Probably you just need to rephrase slightly.

Discussion – We rephrase this sentence.

This significant finding indicates that Ti0.33Nb0.34Hf0.08Zr0.14Ta0.11 exhibits one of the highest μ0Hc2 in the family of type-A HEA superconductors  

Reviewer 2 Report

In this work, the structural and physical properties of the Ti0.33Nb0.34Hf0.08Zr0.14Ta0.11 HEA were investigated. The HEA demonstrated a critical temperature close to 7.5 K and an upper critical field of 13.5 T. Overall, interesting findings were observed, and the manuscript is well organized and discussed. Following questions should be resolved before publication:

1)    It is better to include the “at.%” at the first appearance of the Ti0.33Nb0.34Hf0.08Zr0.14Ta0.11 HEA in abstract and the main text.

2)    What’s the difference between the type-II superconductor in abstract and the type-A superconductor in the main text?

3)    What’s the size of the arc-melted ingot? In literature, it is usually to melt more than four times to guarantee the homogeneity of the alloy. It is also suggested to characterize the microstructure of the alloy to show the homogeneity.

4)    The authors stated that “Ti0.33Nb0.34Hf0.08Zr0.14Ta0.11 306 exhibits the highest μ0Hc2 in the family of type-A HEA superconductors”, however, the authors only included one HEA superconductor in literature, i.e., the Ta0.33Nb0.34Hf0.08Zr0.14Ti0.11 (at.%). If the present results were only compared with this HEA, such statement should be revised more accurately.

5)    It is better to compare the physical properties with conventional superconductors, especially the alloy superconductor (HEA superconductor). Otherwise, the significance of the work would be degraded.

Author Response

Letter of response

We thank very much referee for carefully reading the paper and for numerous comments. In all cases they were taken into account.

In this work, the structural and physical properties of the Ti0.33Nb0.34Hf0.08Zr0.14Ta0.11 HEA were investigated. The HEA demonstrated a critical temperature close to 7.5 K and an upper critical field of 13.5 T. Overall, interesting findings were observed, and the manuscript is well organized and discussed. Following questions should be resolved before publication:

1)    It is better to include the “at.%” at the first appearance of the Ti0.33Nb0.34Hf0.08Zr0.14Ta0.11 HEA in abstract and the main text.

Discussion – Corrected

2)    What’s the difference between the type-II superconductor in abstract and the type-A superconductor in the main text?

Discussion – Type-I and type-II terms are connected with the general classification of superconductors without and with the vortex state, respectively. Additionally, Sun and Cava proposed the classification of HEA superconductors. As it is mentioned in the Introduction, according to their work, our HEA can be classified as type-A HEA superconductor.

3)    What’s the size of the arc-melted ingot? In literature, it is usually to melt more than four times to guarantee the homogeneity of the alloy. It is also suggested to characterize the microstructure of the alloy to show the homogeneity.

Discussion – In the revised manuscript it is added that “the ingot with mass of approximately 1 g was arc-melted at least four times to improve the homogeneity of the alloy.” The microstructure of the alloy was characterized by SEM-EDXS measurements which are presented in the manuscript and Supplementary Materials.

4)    The authors stated that “Ti0.33Nb0.34Hf0.08Zr0.14Ta0.11 306 exhibits the highest μ0Hc2 in the family of type-A HEA superconductors”, however, the authors only included one HEA superconductor in literature, i.e., the Ta0.33Nb0.34Hf0.08Zr0.14Ti0.11 (at.%). If the present results were only compared with this HEA, such statement should be revised more accurately.

Discussion – In the revised manuscript we compare the superconducting properties of the studied HEA with many other type-A HEA superconductors (see Ref. 5-12, 17).

5)    It is better to compare the physical properties with conventional superconductors, especially the alloy superconductor (HEA superconductor). Otherwise, the significance of the work would be degraded.

Discussion – In the revised manuscript we compare the superconducting properties of the studied HEA, especially Hc2, with conventional NbTi and Nb3Sn superconductors.

Reviewer 3 Report

The reviewed article presents the synthesis and detailed investigation of the superconductivity in titanium and niobium-rich new type-A HEA superconductor.

Here are some suggestions for improving the article:

Introduction: It is advisable to summarize the research advancements made in the Ta-Nb-Hf-Zr-Ti system over the past 5 years. Or please write more about the same two-three phase systems without alloying.

Please cross-check the data in Ref. 5 and verify the chemical composition studied, aligning it with the information in line 21.

In line 42, please reference the relevant articles containing the data.

Line 220. Add K for the temperature data.

Additionally, please add information on the sample sizes used for investigating the magnetic properties.

From lines 289 to 298. Are there any comparisons between similar calculations and experimental data on less alloyed materials, not HEA?
What does the "
cocktail effect" mean for these alloys and how is it expressed in the materials? What factors does it have?

Author Response

Letter of response

We thank very much referee for carefully reading the paper and for numerous comments. In all cases they were taken into account.

The reviewed article presents the synthesis and detailed investigation of the superconductivity in titanium and niobium-rich new type-A HEA superconductor.

Here are some suggestions for improving the article:

Introduction: It is advisable to summarize the research advancements made in the Ta-Nb-

Hf-Zr-Ti system over the past 5 years. Or please write more about the same two-three phase

systems without alloying.

Discussion – In the revised manuscript we have added information on the superconducting properties of ternary Nb-Ti-Hf and Nb-Ti-Ta and quaternary Nb-Ti-Ta-Hf alloys.

Please cross-check the data in Ref. 5 and verify the chemical composition studied, aligning

it with the information in line 21.

Discussion – Corrected.

In line 42, please reference the relevant articles containing the data.

Discussion – Corrected.

Line 220. Add K for the temperature data.

Discussion – Corrected.

Additionally, please add information on the sample sizes used for investigating the

magnetic properties.

Discussion – The information on the sample size used for investigating the magnetic properties is added in the revised manuscript.

From lines 289 to 298. Are there any comparisons between similar calculations and

experimental data on less alloyed materials, not HEA?

Discussion – In the revised manuscript it has been added that similar problems with reproducing experimental Tc using theoretical calculations have also been reported in the literature for various materials such as Nb3Ge, V or MgCNi3.

What does the "cocktail effect" mean for these alloys and how is it expressed in the

materials? What factors does it have?

Discussion – For metallic alloys, the cocktail effect indicates that the unexpected properties can be obtained after mixing many elements, which could not be obtained from any single element independently. We do not know what factors play a crucial role in this system, so we have noted that the explanation for the discrepancy between theory and experiment remains unknown.

Reviewer 4 Report

The paper by Idczak et al. presents an intriguing high entropy alloy (HEA) superconductor with Tc=7.5K. They effectively enhanced the upper critical field to 13.5 T, surpassing other alloys with less than 11 T by increasing the Ti percentage. The authors utilized multiple techniques to characterize the properties of this alloy, which holds great promise for future applications in high-field magnets.

However, there are some aspects that need attention before suggesting publication in Materials:

1. The growth procedures and recipes were not clearly explained, making it challenging for other researchers to replicate the results. This information is fundamental to the entire paper.

2. Considering the upper critical field is only 13.5 T, which is within the range of common 14T magnets used in research, it is recommended for the authors to collaborate with a group that can directly measure with 14T magnet. This would provide more solid evidence than fitting with the Ginzburg-Landau (GL) equation.

3. Why the transition in MT is much broader than that in RT? Can the authors explain that?

4. To me, the onset Tc in 9T is not at 3K. It's more like an artificial effect than a transition. And the claim in Line 150 is not solid.

Minor errors:

1. The abbreviation "HEA" should not appear in the abstract without its full expression.

2. The authors have redundant expressions in the paper, like the repetition of the γ value in Line 215, which was already mentioned in Line 209.

3. The authors should use abbreviations for the chemical formula of the material only within the appropriate context.

Author Response

Letter of response

We thank very much referee for carefully reading the paper and for numerous comments. In all cases they were taken into account.

The paper by Idczak et al. presents an intriguing high entropy alloy (HEA) superconductor

with Tc=7.5K. They effectively enhanced the upper critical field to 13.5 T, surpassing other

alloys with less than 11 T by increasing the Ti percentage. The authors utilized multiple

techniques to characterize the properties of this alloy, which holds great promise for future

applications in high-field magnets.

However, there are some aspects that need attention before suggesting publication in

Materials:

  1. The growth procedures and recipes were not clearly explained, making it challenging for

other researchers to replicate the results. This information is fundamental to the entire

paper.

Discussion – In the revised manuscript we present all information which are necessary to replicate the results.

  1. Considering the upper critical field is only 13.5 T, which is within the range of common

14T magnets used in research, it is recommended for the authors to collaborate with a

group that can directly measure with 14T magnet. This would provide more solid evidence

than fitting with the Ginzburg-Landau (GL) equation.

Discussion – In the revised manuscript we propose WHH model which is more adequate for description of Hc(T) data.  Unfortunately at this moment we cannot perform measurements above 9T.

  1. Why the transition in MT is much broader than that in RT? Can the authors explain that?

Discussion – In general, the width of superconducting transition observed in many materials by magnetic measurements is often broader than for resistivity. This effect is mainly connected with the fact that magnetization measured below Tc is proportional to the superconducting volume fraction while the resistivity is not. Formation of one stable superconducting path in the studied material below Tc is enough to observe a sharp drop in resistivity. In the case of HEA, we have an additional effect connected with the high degree of atomic disorder in HEA. For example, when T increases from below towards Tc, some nanoscale regions of HEA with locally lower Tc undergoes transition into the normal state. In result, the magnetic susceptibility and magnetization increase when the superconducting volume fraction decreases with T. At the same time, when the superconducting volume fraction is for example 50% or 25% the resistivity measurements still show zero-resistance state.

  1. To me, the onset Tc in 9T is not at 3K. It's more like an artificial effect than a transition.

And the claim in Line 150 is not solid.

Discussion – We agree that the transition in ac magnetic data at 9T looks like an artificial effect (Fig. 4). However, this result in confirmed by resistivity and magnetization measurements (Fig. 3 and 5).

Minor errors:

  1. The abbreviation "HEA" should not appear in the abstract without its full expression.
  2. The authors have redundant expressions in the paper, like the repetition of the γ value in

Line 215, which was already mentioned in Line 209.

  1. The authors should use abbreviations for the chemical formula of the material only

within the appropriate context.

Discussion – All minor errors are corrected in the revised manuscript.

Round 2

Reviewer 1 Report

Let me start by commending the authors on taking the reviewers' comments seriously and improving the manuscript. I believe that just a few typing or grammatical errors need to be corrected and the article can be published.

- line 1
This is a better word order:
"the new titanium- and niobium-rich type-A high-entropy alloy superconductor"
(Type-A applies to HEA superconductors and not superconductors in general.)

- Just a quick find and replace needed over the whole article. It should be "type-A HEA superconductor" with a hyphen as in this case "type-A" is a compound adjective and should be hyphenated. This will also improve the readability.

- line 21
Wrong reference. Should be DOI: 10.1103/PhysRevApplied.10.014030.

- line 252
"THE normal" should be "the normal".

-line 355
13.5(3) T should be 12.2 T plus error estimate.

- supplement line 11
Each of those atoms was then DISPLACED in all directions

Author Response

Once more, we thank very much referee for carefully reading the paper and for numerous comments. In all cases they were taken into account.

  • - line 1

This is a better word order: "the new titanium- and niobium-rich type-A high-entropy alloy superconductor" (Type-A applies to HEA superconductors and not superconductors in general.)

Discussion - Corrected

  • - Just a quick find and replace needed over the whole article. It should be "type-A HEA superconductor" with a hyphen as in this case "type-A" is a compound adjective and should be hyphenated. This will also improve the readability.

Discussion - Corrected

  • - line 21

Wrong reference. Should be DOI: 10.1103/PhysRevApplied.10.014030.

Discussion - Corrected

  • - line 252

"THE normal" should be "the normal".

Discussion - Corrected

  • -line 355

13.5(3) T should be 12.2 T plus error estimate.

Discussion - Corrected

  • - supplement line 11

Each of those atoms was then DISPLACED in all directions

Discussion - Corrected

Reviewer 2 Report

The manuscript was improved. The reviewer still has one concern before publication. As mentioned in comment 2, "our HEA can be classified as type-A HEA superconductor", why in abstract, it was classified as the type II superconductor? While in conclusion, it was classfied as type 
A superconductor.  It was confused for readers. It is better to describe the classifications more clearly in the manuscript. 

Author Response

Once more, we thank very much referee for carefully reading the paper and for comment.

The manuscript was improved. The reviewer still has one concern before publication. As mentioned in comment 2, "our HEA can be classified as type-A HEA superconductor", why in abstract, it was classified as the type II superconductor? While in conclusion, it was classfied as type A superconductor.  It was confused for readers. It is better to describe the classifications more clearly in the manuscript. 

Discussion – Introduction and Conclusions are corrected according to this comment.

Reviewer 4 Report

I appreciate the authors' efforts in the revised manuscript. My questions have been addressed. I would like to suggest the publication in the Materials. 

One more thing the authors need to check is the curve plotting in Fig.4 for B = 0.1 T and 0.5T. The plotting lines jump back and forth at some data points. 

Author Response

Once more, we thank very much referee for carefully reading the paper and for comment.

I appreciate the authors' efforts in the revised manuscript. My questions have been addressed. I would like to suggest the publication in the Materials.

One more thing the authors need to check is the curve plotting in Fig.4 for B = 0.1 T and 0.5T. The plotting lines jump back and forth at some data points.

Discussion – We correct the curve plotting in Fig. 4.
